# Single-Step Process for Titanium Surface Micro- and Nano-Structuring and In Situ Silver Nanoparticles Formation by Ultra-Short Laser Patterning

**DOI:** 10.3390/ma15134670

**Published:** 2022-07-03

**Authors:** Dante Maria Aceti, Emil Filipov, Liliya Angelova, Lamborghini Sotelo, Tommaso Fontanot, Peyman Yousefi, Silke Christiansen, Gerd Leuchs, Stanislav Stanimirov, Anton Trifonov, Ivan Buchvarov, Albena Daskalova

**Affiliations:** 1Institute of Electronics, Bulgarian Academy of Sciences, 72 Tzarigradsko Chaussee Blvd., 1784 Sofia, Bulgaria; emil.filipov95@gmail.com (E.F.); lily1986@abv.bg (L.A.); albdaskalova@gmail.com (A.D.); 2Department of Physics, Friedrich-Alexander-Universität Erlangen-Nürnberg, Staudtstraße 7, 91058 Erlangen, Germany; sotelo.lamborghini@gmail.com (L.S.); pn.yousefi@gmail.com (P.Y.); gerd.leuchs@mpl.mpg.de (G.L.); 3Innovations-Institut für Nanotechnologie und Korrelative Mikroskopie gGmbH Äußere Nürnberger Str. 62, 91301 Forchheim, Germany; silke.christiansen@ikts.fraunhofer.de; 4Fraunhofer Institute for Ceramic Technologies and Systems IKTS Äußere Nürnberger Str. 62, 91301 Forchheim, Germany; tommaso.fontanot@ikts.fraunhofer.de; 5Max-Planck-Institut für die Physik des Lichts, 91058 Erlangen, Germany; 6Faculty of Chemistry and Pharmacy, Sofia University, 1 J. Bourchier Blvd., 1164 Sofia, Bulgaria; sstanimirov@chem.uni-sofia.bg; 7Department of Physics, Sofia University, 5 J. Bourchier Blvd., 1164 Sofia, Bulgaria; a.trifonov@phys.uni-sofia.bg (A.T.); ivan.buchvarov@phys.uni-sofia.bg (I.B.)

**Keywords:** ultra-short laser processing, titanium, silver nanoparticles, surface patterning, laser ablation, multiphoton photo-reduction

## Abstract

Ultra-short laser (USL)-induced surface structuring combined with nanoparticles synthesis by multiphoton photoreduction represents a novel single-step approach for commercially pure titanium (cp-Ti) surface enhancement. Such a combination leads to the formation of distinct topographical features covered by nanoparticles. The USL processing of cp-Ti in an aqueous solution of silver nitrate (AgNO_3_) induces the formation of micron-sized spikes surmounted by silver nanoparticles (AgNPs). The proposed approach combines the structuring and oxidation of the Ti surface and the synthesis of AgNPs in a one-step process, without the use of additional chemicals or a complex apparatus. Such a process is easy to implement, versatile and sustainable compared to alternative methodologies capable of obtaining comparable results. Antimicrobial surfaces on medical devices (e.g., surgical tools or implants), for which titanium is widely used, can be realized due to the simultaneous presence of AgNPs and micro/nano-structured surface topography. The processed surfaces were examined by means of a scanning electron microscope (SEM), energy-dispersive X-ray spectroscopy (EDX), atomic force microscopy (AFM) and Raman spectroscopy. The surface morphology and the oxidation, quality and quantity of AgNPs were analyzed in relation to process parameters (laser scanning speed and AgNO_3_ concentration), as well as the effect of AgNPs on the Raman signal of Titanium oxide.

## 1. Introduction

Surface features and properties of different biocompatible materials are crucial for a huge variety of applications, especially in the field of tissue engineering and regenerative medicine. Ultra-short laser processing represents a versatile tool whose recognition in this field has been increasing [1,2]. The use of ultra-short pulse laser radiation for surface structuring has been widely explored in material surface processing [3], as well as in the synthesis of metallic nanoparticles (NPs) by photoreduction [4,5]. Laser patterning is demonstrated to be an effective tool to engineer any kind of material surface (e.g., metals, polymers, ceramics, etc.) [6], tailoring their topographic properties in accordance with the specific biomedical application. In fact, it makes it possible to tune the morphological features, modify the interface properties, such as wettability [7,8], wear-resistance and friction [9], or provide optical properties such as structural color [10]. Moreover, the use of ultra-short pulses offers several further advantages: higher precision, less debris formation and reduced thermal damage [11,12]; moreover, the extremely high spatial-temporal density of energy provided by such laser systems is necessary for a multiphoton absorption and to trigger the correlated phenomena [13] (e.g., photopolymerization, photoreduction, fluorescence, etc.). Metallic NPs are of great interest, finding application in several fields, such as photonics [14], catalysis [15], bio-medicine [16,17], etc. The in situ synthesis of metallic NPs, due to femtosecond laser irradiation, has opened new possibilities ranging from surface patterning [18,19] to additive manufacturing [20,21]. The exploitation of multiphoton photoreduction allows for the spatial confinement of the synthesis of nanoparticles [22]. In such a way, the synthesis confinement entails a reduced number of synthesized particles.

The ultra-short laser processing of commercially pure Titanium (cp-Ti) immersed in an aqueous solution of silver nitrate (AgNO_3_), shown in Figure 1, represents a novel approach not yet explored in the literature. It allows for the combination of surface structuring and the in situ synthesis of nanoparticles, which leads to the formation of micro- and nano-sized topographical features surmounted by silver nanoparticles (AgNPs). Furthermore, the process induces the oxidation of titanium, leading to the formation of surface features covered by a layer of titanium oxide (see Figure 1c).

Titanium surface structuring and AgNPs addition are of great interest for application in regenerative medicine. Titanium is one of the most used materials in medical implants, and its surface morphology and oxidation have a great impact on cell behavior [23,24,25,26,27] after surgery, especially on cell adhesion (the topography could resemble the natural tissue microarchitecture and roughness) and orientation (directing the cells in the desired direction) and, hence, on proliferation and differentiation as subsequent steps in the tissue formation process [28,29]. On the other hand, silver, in the form of ions or nanoparticles, is one of the most effective antimicrobial agents [30]. The overall surface topography can effectively prevent bacterial adhesion [31,32,33] by reducing the contact points or by exerting mechanical pressure [34] on the bacterial cells, which could lead to cell rupture. Laser surface texturing represents a promising method, as an alternative to coatings, to realize antimicrobial surfaces [35,36] by modifying their topography and chemical properties. Schwibbert et al. [37], for example, studied the antimicrobial efficacy of laser-modified polyethylene. The use of AgNPs to ensure antibacterial properties for a variety of bio-compatible materials has been widely explored [38], finding various biomedical applications [39]. The group of Thomas et al. [40] has experimentally demonstrated that the AgNPs-coated catheter surfaces are effective against *S.aureus* biofilm formation. In another study, Lee et al. [41] incorporated AgNPs in orthodontic resin, which showed strong antimicrobial activity against oral pathogens. In the research of Mirzaee et al. [42], it was observed how the silver doping of a hydroxyapatite coating ensures better wettability and corrosion resistance.

The combined presence of Titanium oxide and AgNPs was demonstrated to positively affect material performances in the biological environment [43]. Moreover, Arens et al. [44] have recently demonstrated the in vivo biocompatibility of implants coated by a silver-enriched oxide layer. Different approaches have been explored to obtain a structured titanium surface functionalized with AgNPs, employing different techniques [45,46] or complex apparatuses [47]. Surface modification and particle synthesis were obtained by Lin et al. [48] on top of an optical fiber for SERS (surface-enhanced Raman scattering) applications, while Yin et al. [49], Han et al. [50] and Youfu et al. [51], achieved similar results, combining surface modification and AgNPs deposition but performing the two processes separately, in two consecutive steps. The silver coatings of titanium were obtained by Ewald et al. [52] and Bai et al. [53] by means of physical vapor deposition and magnetron sputtering, respectively. Such surfaces showed antimicrobial activity while maintaining good cytocompatibility. Zhao et al. [54] obtained an effective antibacterial coating by incorporating AgNPs in Titania, nanostructured via electrochemical anodization. The particle inclusion was obtained by immersion in AgNO_3_ and UV irradiation. Other approaches to functionalize the Ti surface with AgNPs and improve its biological and antimicrobial properties involve the chemical synthesis of the particles and additional processes for their deposition [55,56].

Such results demonstrate that the use of Ag-coated Ti appears promising for biomedical applications (e.g., the surface treatment of prostheses and other medical tools). In this context, the exploitation of laser processing appears to have a potential that has not yet been fully exploited. Laser patterning offers the possibility to obtain coating and surface structuring without the need for chemical processes. Moreover, it provides the possibility of customized solutions, in terms of material shape and size (no need for a vacuum chamber, flat surfaces, etc.) and the tunability of the process while scanning the surface. Each portion of surface area can be treated differently. Nevertheless, the laser patterning of surfaces with AgNPs for such applications has not been widely explored in the current state of the literature. Femtosecond laser surface modification combined with the fabrication of silver nanostructures was demonstrated by MacKenzie et al. [57]; however, the two processes were carried out separately. The possibility of obtaining surface structuring and coating by means of laser treatment was demonstrated by Zhao et al. [58]; this was made using several consecutive steps to achieve such result.

The method described in this paper represents a simple and sustainable one-step alternative approach for material surface treatment for biomedical applications, since the USL processing induces the structuring and oxidation of the surface. Thus, the use of chemicals is no longer necessary. Similarly, the synthesis of AgNPs by photoreduction does not require additional reagents or stabilizing agents [59], reducing the waste of material, since the synthesis is localized directly on the surface, and also preventing the contamination by unwanted chemical agents. We demonstrated the possibility offered by our simple and easy approach to modify the surface morphology, up to the micro-scale, while decorating the obtained features with AgNPs. We evaluated the influence of processing parameters (i.e., scanning speed) and process conditions (i.e., AgNO_3_ concentration in a solution).

## 2. Materials and Methods

Commercially pure Titanium plates (Sigma Aldrich, 10 mm × 10 mm × 2 mm) placed in a cylindrical container (with a 55 mm diameter) and immersed in an aqueous solution with rising concentrations (0.0, 0.6, 1.2 and 6 mM or 0.0, 0.01, 0.02 and 0.1 wt.%) of Silver Nitrate (AgNO_3_) were patterned using a Ti:Sapphire laser (Quantronix-Integra-C, Hamden, CT, USA) with a pulse duration τ < 130 fs, a central wavelength λ = 800 nm and a repetition rate ω = 1 kHz. The experimental setup and condition are schematically shown in Figure 2.

The laser beam, 7 mm in diameter, was focused on the sample surface through 3 mm of the solution by an achromatic biconvex lens with a 100 mm focal distance. The focal spot diameter is d~50 μm. The laser power was fixed, by means of a neutral density filter, to a value of P = 120 mW, corresponding to a fluence of F = 15.3 J/cm^2^, and a two-axis motorized software-controlled (LabView) translation stage (Thorlabs, Newton, New Jersey, United States) allowed for the movement of the sample, setting different values of scanning speed (V = 0.1, 0.2, 0.35, 0.5 mm/s) with a fixed hatch distance (100 μm). Patterns with the 16 different combinations of parameters were obtained. After laser processing, the material was rinsed with deionized water (d.i.H_2_O) and ethanol (70%).

The morphology and chemical composition of the sample surface after the laser treatment were studied. The influence of the processing parameters and AgNO_3_ concentration on the surface features alongside the oxide formation, quantity and quality of AgNPs was evaluated. A scanning electron microscope (SEM, Tescan Lyra 3 GMU, Brno, Czech Republic) was used to acquire images of the sample surface with different magnifications using a 5 kV voltage and a Secondary Electron (SE) detector. Particle size measurement was performed by analyzing the SEM images via software (ImageJ).

An EDX detector (Ultim Max, Oxford Instruments, Abingdon on Thames, UK) mounted on a Crossbeam 550 SEM (Carl Zeiss Microscopy, Deutschland GmbH, Germany) was used to acquire spectra and elemental distribution maps using a 10 kV electron beam voltage and 40 kx magnification. The values reported were obtained as the average of three acquisitions.

An atomic force microscope (AFM, Park System, NX20, Suwon, South Korea) was used to scan the surface of the samples, acquiring 20 μm × 20 μm 2D frames, in order to study the morphology of the sample surface and evaluate its roughness. Four different areas were studied on each pattern, and the resulting values of roughness (Sa, arithmetical mean height) were averaged.

Raman spectra (spectrometer Horiba MicroRaman, Kyoto, Japan) were obtained using a 532 nm laser with a power *p* = 1.68 mW in the absence of silver NPs and *p* = 52.5 μW in the presence of silver nanoparticles.

## 3. Results and Discussion

The SEM images of the laser-processed (fluence F = 15.3 J/cm^2^, scanning speed V = 0.1, 0.2, 0.35 and 0.5 mm/s) cp-Ti surface immersed in a 0, 0.01, 0.02 and 0.1 wt.% AgNO_3_ presented in Figure 3 revealed a structured surface topography consisting of an alternation of micrometric peaks and even submicrometric valleys. The samples processed in the presence of AgNO_3_ were characterized by the AgNPs distribution mainly on the top of the created peaks, while the pits appeared mostly free of nanoparticles.

The results indicated that a lower laser scanning speed, corresponding to a higher number of laser pulses deposited per unit length, led to the formation of peaks with larger dimensions on the titanium surface and to the greater presence of silver particles (Figure 3i–p).

The laser processing of the Ti samples in 0.1 wt.% AgNO_3_ or with V = 0.1 mm/s yielded the formation of particles with dimensions even greater than 500 nm in diameter on the surface of the Ti samples. Only in the case of 0.01 wt.% AgNO_3_ and V = 0.5 mm/s (Figure 3h) were most of the particles below 100 nm in size and, in any case, not over 120 μm (see Table 1).

The results from the performed EDX analysis revealed that the combination of higher concentrations of AgNO_3_ with lower scanning speeds (0.1 wt.% AgNO_3_, V = 0.1 mm/s) led to a sensitively higher concentration of AgNPs on the surface of the laser-structured Ti samples (as can be seen from Figure 4a).

The EDX elemental composition ratio (wt.%) between the oxygen and titanium O (wt.%)/Ti (wt.%) is presented in Figure 4b. By increasing the scanning speed, the amount of detected [O] seems to increase as well, which could be attributed to the further surface oxidation of the material. However, even if the trend appears quite clear, the increase remains comparable to the value deviation (see error bars). On the other hand, the higher dose of energy (i.e., lower V) introduced at the site of laser–material interaction leads to deeper and stronger etching of the surface and a greater material ejection instead of its oxidation.

The EDX elemental distribution maps presented in Figure 5 are in accordance with the results from the performed SEM analysis. As can be clearly seen from the figure, the AgNPs are mainly located on the top of the surface features (see Figure 5c), while no particle formation could be seen in the pits. The presence of spherical titanium oxide particles, alongside the vast majority of AgNPs, was also observed (see Figure 5d). The discrimination between the two kinds of particles was evident when comparing the secondary electron (SE) image with the elemental distribution maps (titanium oxide particles are indicated by white arrows). Figure 5d shows that the surface features were rich in titanium oxide, unlike cavities, where the only detected signal originated from pure titanium.

The AFM images presented in Figure 6 further supported the observations that by lowering the translation stage scanning speed (V = 0.10 and 0.20 mm/s), the laser-induced microstructures on the surface of the Ti samples tended to merge with each other, becoming larger in size (see Figure 6b). Contrariwise, a higher scanning speed (V = 0.35 mm/s, 0.50 mm/s) ensured smaller surface microgranular structures, as can be noticed from Figure 6a.

The increase in surface microroughness after the laser processing could be mostly attributed to the formed micro-structures on the metal surface but also to the presence of AgNPs (Figure 7). The main trend in the obtained Sa values suggested that the presence of AgNO_3_ led to a smoother surface. This observation was more evident when the Sa of the surface processed at a low scanning speed (0.1–0.35 mm/s) and low AgNO_3_ concentration (0.01–0.02 wt.%) was compared with the one processed in deionized water.

The energy required for the photo-reduction of silver ions (Ag^+^) was deducted from the total energy that was required for the structuring of Ti in the absence of AgNO_3_. This partial uptake of the total laser energy by AgNO_3_ resulted in a lower amount of energy absorbed by the metal; thus, a decrease in surface roughness was observed. However, such an effect at high concentrations of AgNO_3_ could be compensated by the formation of micron-sized silver particles and clusters, which provided a significant contribution to the overall surface roughness.

The oxide layer formed on the top of the peaks as a result of the laser–metal interaction was investigated by Raman spectroscopy. In both cases of cp-Ti processed in deionized water (Figure 6a) and immersed in the AgNO_3_ solution (Figure 6b), the oxide layer produced a broad signal, in which it was not possible to clearly distinguish between the specific peaks and the contributions of the various oxides that titanium could form. According to Landis et al. [60], a major contribution in the Raman spectra presented in both figures is expected to be due to titanium dioxide (TiO_2_) in a low crystallinity to an amorphous state; the characteristic peaks of Anatase and Rutile cannot be identified. However, the presence of different oxides, such as Ti_2_O_3_, cannot be excluded.

The observed signal of the oxide in the presence of silver particles (Figure 8b) was remarkably more intense compared to the signal obtained from the cp-Ti processed in deionized water. In order to avoid detector saturation, it was necessary to significantly reduce the laser power from 1.68 mW to 52.5 μW (which means a reduction by a factor of 32). In similar conditions, the sample without silver particles would have produced a signal indistinguishable from noise. Moreover, the presence of silver particles made evident otherwise undetectable modes at ~750 cm^−1^ and in the range 1200–1700 cm^−1^. The enhancement of Raman scattering was reduced when probing the sample using a 785 nm laser, which does not match the plasmon resonance of AgNPs [61]. Such observations confirmed that the oxide signal was enhanced by the SERS (Surface Enhanced Raman Scattering) effect induced by AgNPs.

The use of ultra-short laser radiation opens a wide variety of possibilities in surface structuring with feature dimensions even below the diffraction limit, exploiting various effects arising from the interaction with different materials. Such structures are generally identified as LIPSS (Laser Induced Periodic Surface Structures). Under this definition, several morphologies are collected and characterized by different sizes, shapes, symmetries, etc. (ripples, grooves, spikes and their combinations [62,63,64]). All such characteristics arise from complex dynamics [65] originating from the ultra-short laser–matter interaction: the influence of laser parameters (e.g., wavelength, pulse duration, power, the number of pulses, the angle of incidence, etc.) and material properties (e.g., refractive index, dielectric constant, viscosity, ablation threshold, etc.).

The absence of periodic and oriented ripples with sub-wavelength periodicity expected to cover the topographical features [66] obtained in the current study can be explained by the bubble formation occurring at the metal–liquid interface during the laser processing as a result of the high temperature reached in the focal spot. The formation and rising of the bubbles interfere with the focusing and locally distort the polarization of the light. This makes the deposition of several identical pulses impossible, which is a necessary condition to induce the formation of such kinds of ripples. The material ejection, the presence of particles and the turbulent motion into the fluid disturb the beam shape, which contributes to the absence of periodic and oriented ripples on the sample’s surface. Koch et al. [67] also verified the absence of periodic ordered ripples on laser-processed material surfaces while immersed in water. However, as demonstrated by Shen et al. [68] and Rivera et al. [69], it is also possible to obtain such a structuring (subwavelength periodic ripples) by laser irradiation under water, but its combination with microstructures cannot yet be directly obtained in one step.

The combination of surface structuring by laser ablation and NPs synthesis by multiphoton photoreduction allows for the formation of unique topography-alternating pits and AgNPs-covered peaks. Moreover, the particle adhesion is promoted, as their formation takes place simultaneously with surface reconstruction, The proof is the absence of particles inside the pits (as shown by the SEM and EDX analyses), where most of the material ejection takes place, during the ablation process, and such flow of matter prevents the deposition of silver particles. This results in the further confinement of AgNPs only above the oxide layer on the micro-peaks. Since the particles are not covering the surface homogeneously, being localized only on top of the micro-peaks, the total amount of silver deposited is reduced and the waste of material is avoided.

The methodology presented in this paper, as already highlighted, could represent a novelty in the treatment of biomedical materials. Its facile implementation, as well as the absence of pre- or post-treatments, make the process easy to handle and to scale up for material surface treatments. Moreover, the absence of chemical reagents not only facilitates the process but reduces the possibility of contamination. All of this makes it a versatile process to be exploited in the creation of antimicrobial surfaces, representing a convenient alternative to other methodologies that require several steps for sample preparation and the use of complex apparatuses which often involve limitations in terms of the material, size and shape of the sample to be processed. Furthermore, the USL processing to produce rough surfaces capable of exploiting the SERS effect due to the presence of plasmonic materials has aroused growing interest in recent years [70,71]. This opens the possibility, for our process, to find an application in the realization of SERS-based biosensors. Similar structures could also find an application for Laser Induced Breakdown Spectroscopy (LIBS), especially in Nanoparticle Enhanced LIBS (NELIBS) [72], which still represents a developing field [73]. Surface patterning with metal NPs also allows for the exploitation of the thermoplasmonic effect, opening another wide range of potential applications [74,75] for our method which provide, in addition, surface structuring. However, finer control in the NPs’ size and distribution is required to ensure an optimal match between the NPs’ plasmonic resonance and the used laser wavelength, a necessary condition for the aforementioned applications. Further experiments and investigations will explore the potentiality of our method for such applications as well.

## 4. Conclusions

The one-step process proposed in this paper, which involves the ultra-short pulse laser processing of cp-Ti samples in a solution (deionized water or increasing concentrations of AgNO_3_), combines additive and subtractive manufacturing, which allows for the obtention of a microstructured titanium surface with AgNPs deposited on the top. The ultra-short laser processing of titanium in an aqueous solution of AgNO_3_ not only induces surface structuring and oxidation, but it further promotes the formation and deposition of AgNPs onto the material by direct photoreduction. Moreover, the degree of surface roughness, as well as the amount and size of single AgNPs or clusters, can be easily tailored. In the current study, it has been shown how the AgNO_3_ concentration and laser scanning speed influence the quantity and the size of the obtained AgNPs, as well as the titanium topographical features (e.g., roughness) and oxidation. Additionally, the SERS effect on the titanium oxide signal has been reported due to the presence of AgNPs.

Our process represents a convenient and promising solution for surface structuring and functionalization with nanoparticles which is applicable to a variety of materials. The use of titanium and silver could find an application for antimicrobial surfaces realization, as the overall morphology, presenting micron-sized features, could result in bacteria trapping, while the presence of AgNPs would ensure bactericidal activity. The use of the proposed process as a surface treatment of prosthetic implants could prevent bacteria colonization and its consequences. Moreover, the rough surface, together with the presence of oxide, would play an important role in terms of biocompatibility, and the AgNPs patterning makes use of lower quantities of silver compared to traditional coating techniques and is less likely to have a cytotoxic effect than a homogeneous layer on the surface.

The possibility of achieving titanium structuring and oxidation with no need for mechanical, chemical or thermal treatments, together with the synthesis and deposition of AgNPs, makes our method viable for a variety of applications. Moreover, the absence of additional chemicals (which can represent a source of pollution or contamination) and procedures makes the process bio-friendly and easy to scale up. The disposal of hazardous waste and complex equipment and instrumentation, apart from the laser system, is not necessary.

## Figures and Tables

**Figure 1 materials-15-04670-f001:**
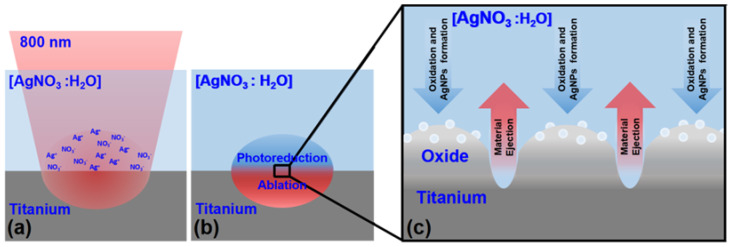
Schematic representation of the main processes involved in the formation of surface characteristics: (**a**) the laser beam is focused on the interface between the Ti surface and AgNO_3_ solution; (**b**) the ultra-short laser radiation induced the ablation of the Ti surface and the photoreduction of silver ions (Ag^+^) in the solution; (**c**) the two processes combined led to the formation of surface features covered by titanium oxide and surmounted by AgNPs.

**Figure 2 materials-15-04670-f002:**
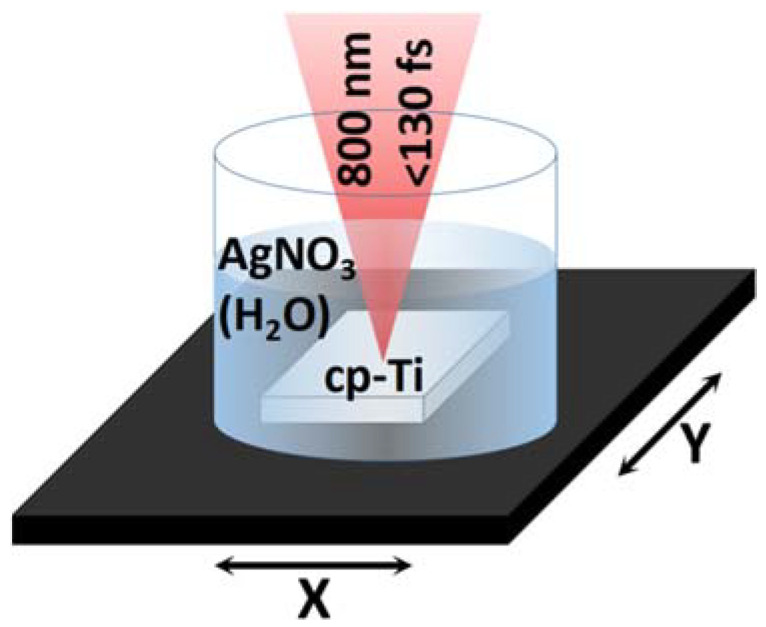
Schematic representation of the ultra-short laser processing of a cp-Ti plate fixed on a two-axis motorized software-controlled (LabView) translation stage and immersed in an aqueous solution of silver nitrate (AgNO_3_).

**Figure 3 materials-15-04670-f003:**
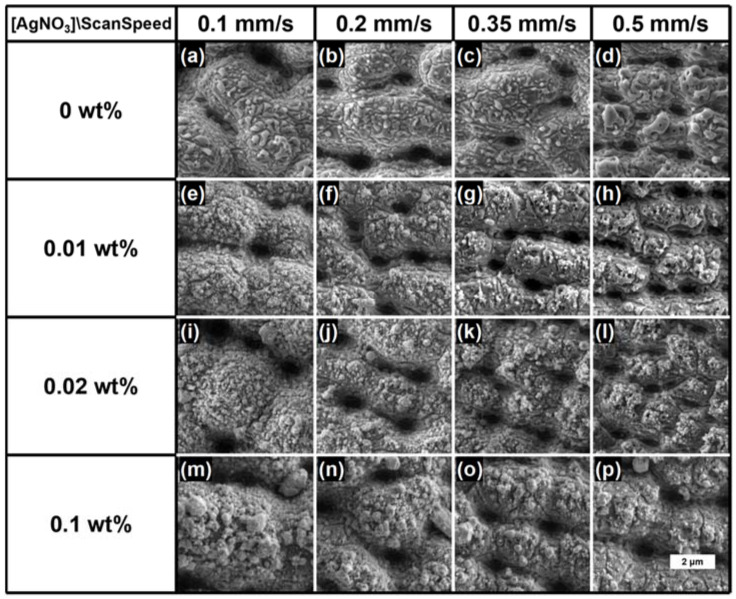
SEM images of the laser-processed cp-Ti surface immersed in AgNO_3_ at different concentrations (0, 0.01, 0.02 and 0.1 wt.%); V = 0.1, 0.2, 0.35 and 0.5 mm/s and F = 15.3 J/cm^2^; scale bar: 2 μm.

**Figure 4 materials-15-04670-f004:**
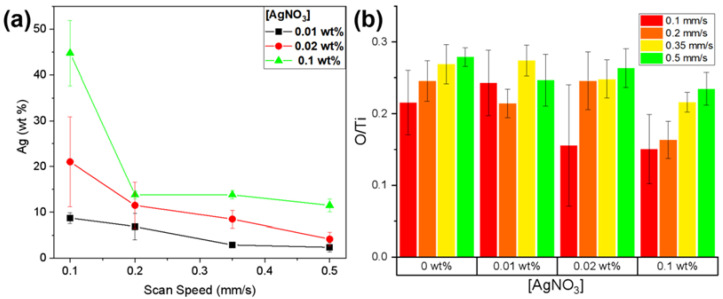
(**a**) Plot of Ag (wt.%) detected by EDX analysis for V = 0.1, 0.2, 0.35, 0.5 mm/s and 0.01, 0.02, 0.1 wt.% AgNO_3_ concentration in the solution; (**b**) O (wt.%)/ Ti (wt.%) EDX elemental composition ratio for each different combination of processing conditions: V = 0.1, 0.2, 0.35, 0.5 mm/s and 0.01, 0.02, 0.1 wt.% AgNO_3_ concentration in the solution.

**Figure 5 materials-15-04670-f005:**
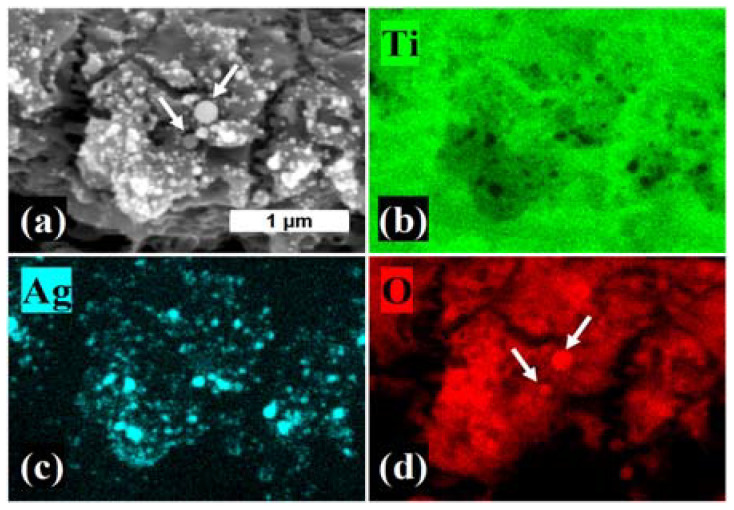
(**a**) A secondary-electron (SE) SEM image of cp-Ti processed in 0.01 wt.% of AgNO_3_ at V = 0.35 mm/s compared to the EDX elemental distribution maps of: (**b**) Ti, (**c**) Ag and (**d**) O. The white arrows indicate the titanium oxide particles; scale bar = 1 μm.

**Figure 6 materials-15-04670-f006:**
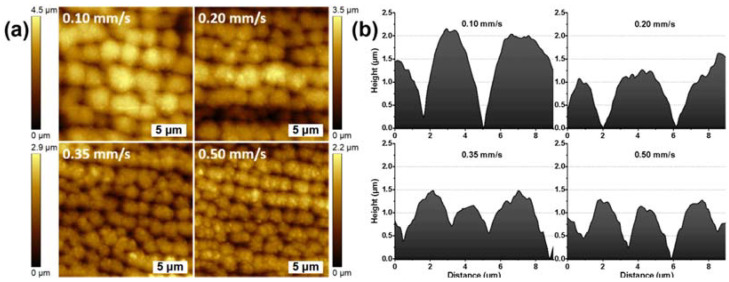
Ti samples processed in pure water with V = 0.1, 0.2, 0.35, 0.5 mm/s; (**a**) 2D AFM images (20 μm × 20 μm); scale bars = 5 μm; (**b**) AFM height profiles.

**Figure 7 materials-15-04670-f007:**
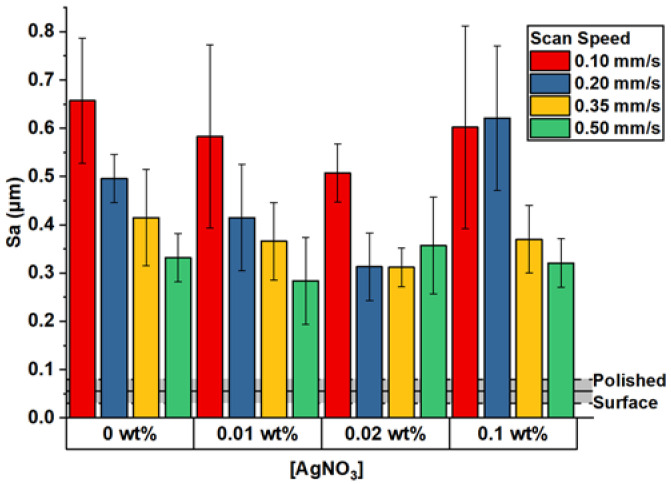
Plot of surface roughness values (Sa) for each different combination of processing conditions: V = 0.1, 0.2, 0.35, 0.5 mm/s and 0.01, 0.02, 0.1 wt.% AgNO_3_ concentration in the solution. The black line represents the original roughness value of the untreated surface and its deviation (gray area).

**Figure 8 materials-15-04670-f008:**
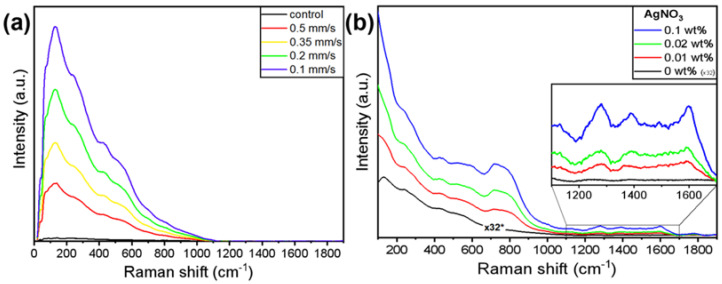
Raman spectra of cp-Ti processed with F = 15.3 J/cm^2^; (**a**) processed in deionized water with V = 0.1, 0.2, 0.35, 0.5 mm/s; all spectra were obtained using a 532 nm laser with a power P = 1.68 mW; the control (black line) is the signal of the untreated material; no evidence of oxide is detected. (**b**) Processed with V = 0.1 mm/s in deionized water (0 wt.% AgNO_3_) and in a solution with 0.01, 0.02, 0.1 wt.% AgNO_3_; all spectra were obtained using a 532 nm laser with P = 52.5 μW in the presence of Ag particles and at P = 1.68 mW in the absence of Ag particles (black line).

**Table 1 materials-15-04670-t001:** Maximum AgNPs diameter measured for V = 0.1, 0.2, 0.35 and 0.5 mm/s; 0, 0.01, 0.02 and 0.1 wt.% AgNO_3_.

Scan Speed	Ag_0.01	Ag_0.02	Ag_0.1
(mm/s)	(μm)	(μm)	(μm)
0.10	0.50 ± 0.01	1.10 ± 0.01	2.00 ± 0.01
0.20	0.40 ± 0.01	0.60 ± 0.01	1.50 ± 0.01
0.35	0.18 ± 0.01	0.50 ± 0.01	1.20 ± 0.01
0.50	0.12 ± 0.01	0.30 ± 0.01	1.00 ± 0.01

## Data Availability

Not applicable.

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
