# Peer review of "Single-Step Process for Titanium Surface Micro- and Nano-Structuring and In Situ Silver Nanoparticles Formation by Ultra-Short Laser Patterning"

_materials, 2022, doi:10.3390/ma15134670_

Round 1
Reviewer 1 Report
In the work of Aceti et al, entitled “Single-Step Process For Titanium Surface Micro and Nano Structuring and In Situ Silver Nanoparticles Formation by Ultrashort Laser Patterning”, the authors demonstrate a one-step procedure for titanium surface with AgNPs deposited on top. The authors study the ease of achieving Ti structuring and simultaneous oxidation and synthesis of AgNPs.
The study is very intriguing and I reckon it has good potential to be published in Materials. However, the manuscript feels to be in an early stage and there are some major issues that need to be addressed before being considered for publication.
1. There are plenty of methods for obtaining AgNPs and oxidized titanium surfaces in the literature and there should be more focus on the novelty of this process. For instance, the schematic representation of Figure 7 in addition to the literature review on Page 9 should be transferred into the Introduction
2. The structural characterization of the produced materials is severely lacking. Let's be sincere, the Raman spectra do not show anything. The anatase and rutile peaks normally are high in intensity whereas in Figure 6 there are no distinctive peaks to confirm the existence of TiO2. Additionally, XRD, which is a fingerprint characterization technique is missing which can confirm the oxidation of cp-Ti.
3. Is the [O] increase presented accurate, as it can be influenced by atmospheric oxygen. If the authors have access to XPS they would be able to provide better evidence of the oxidation.
4. Since in the title the authors mention "Micro and Nano Structures" they should provide height profiles extracted from the AFM analysis.
Overall, while the concept of research looks promising there is not enough data, and the presentation of the research in the manuscript is a bit confusing
Author Response
Response to Reviewer 1 Comments
In the work of Aceti et al, entitled “Single-Step Process For Titanium Surface Micro and Nano Structuring and In Situ Silver Nanoparticles Formation by Ultrashort Laser Patterning”, the authors demonstrate a one-step procedure for titanium surface with AgNPs deposited on top. The authors study the ease of achieving Ti structuring and simultaneous oxidation and synthesis of AgNPs.
The study is very intriguing and I reckon it has good potential to be published in Materials. However, the manuscript feels to be in an early stage and there are some major issues that need to be addressed before being considered for publication.
We want to thank the reviewer for the time and the effort spent to assess our manuscript and for the valuable comments and suggestions provided. We are also glad to know that the reviewer found our study interesting and valid.
Point 1: There are plenty of methods for obtaining AgNPs and oxidized titanium surfaces in the literature and there should be more focus on the novelty of this process. For instance, the schematic representation of Figure 7 in addition to the literature review on Page 9 should be transferred into the Introduction
Response 1: We appreciate the suggestion and we recognize that Figure 7 (now Figure 1) as well as the literature review fits better in the introduction, the picture and the mentioned text is now part of 1.Introduction. Please see page 2 of the revised manuscript, lines 21-23, page 3, lines 27-50, and page 4, lines 1-7.
However, we would like to point out that the novelty lies not in the way of obtaining the structuring and oxidation of titanium or the synthesis of AgNPs. The use of femtosecond lasers to process titanium surfaces or to produce AgNPs by multiphoton photoreduction is well established in literature, but, to the best of our knowledge, there are no examples of the two processes conducted together at the same time. In such a way titanium modification and AgNPs synthesis and deposition are combined in the same single step. Therefore, we believe that such an aspect, representing the uniqueness of our study, is already sufficiently highlighted.
Point 2: The structural characterization of the produced materials is severely lacking. Let's be sincere, the Raman spectra do not show anything. The anatase and rutile peaks normally are high in intensity whereas in Figure 6 there are no distinctive peaks to confirm the existence of TiO2. Additionally, XRD, which is a fingerprint characterization technique is missing which can confirm the oxidation of cp-Ti.
Response 2: We agree with the reviewer that the Raman spectra are not providing enough information to identify which kind of oxide is present. However, the signal itself is proof of oxidation of the titanium. The unprocessed material (identified as “control”, black line, in the Raman spectrum (now Figure 8a), showed only a flat background signal. Broad Raman signals, similar to the one we obtained, are commonly reported for titanium processed by ultrashort laser. We recognize, in agreement with the reviewer, that oxide identification was incorrect. Results from literature (now cited in the manuscript as reference [60]) indicate amorphous TiO2 as the main component of the oxide layer. Please see the corrected paragraph and relative reference:
According to Landis et al.[60], a major contribution in the Raman spectra presented in both figures is expected to be due to titanium dioxide (TiO2) in a low crystallinity to amorphous state; the characteristic peaks of Anatase and Rutile cannot be identified. However the presence of different oxides, such as Ti2O3 cannot be excluded.
- Landis, E.C.; Phillips, K.C.; Mazur, E.; Friend, C.M. Formation of nanostructured TiO2 by femtosecond laser irradiation of titanium in O2. J. Appl. Phys. 2012, 112, 063108.https://doi.org/10.1063/1.4752276
We also agree that XRD would be an effective method to have a better understanding of the nature of the oxide layer. However, unfortunately, we have no direct access to such analysis and the time and resources necessary would exceed our possibilities.
Preliminary XRD analysis, conducted on similarly processed samples, showed difficult to interpret data, due to the lack of crystallinity and the coexistence of different kinds of oxides (e.g. TiO2, Ti2O3, etc.), due to that we have preferred not to invest time and resources in this way.
Point 3. Is the [O] increase presented accurate, as it can be influenced by atmospheric oxygen. If the authors have access to XPS they would be able to provide better evidence of the oxidation.
Response 3. We agree with the reviewer that XPS analysis would provide more and reliable information about titanium and oxygen oxidation states. Unfortunately we cannot easily access this type of analysis. Concerning the detected oxygen by EDX, the analysis is conducted in high vacuum conditions (~10-7 mbar), so the influence of atmospheric oxygen can be excluded. In case the reviewer refers to the processing condition, during the laser treatment, the oxygen for titanium oxidation is taken by the surrounding environment (deionized water in our case), being the sample immersed during the whole procedure, the detected values are only due to differences in process parameters. The unprocessed material shows a complete absence of oxygen (0%).
Point 4: Since in the title the authors mention "Micro and Nano Structures" they should provide height profiles extracted from the AFM analysis.
Response 4: We agree with the reviewer that AFM height profiles provide a more complete picture of the morphology obtained. A new figure showing AFM height profiles (Figure 6b) has been added as the reviewer suggested . Please see the new picture reported below:
*see the attachment for the figure*
Figure 6. Ti samples processed in pure water with V=0.1, 0.2, 0.35, 0.5 mm/s; (a) 2D AFM images (20 μm x 20 μm) scale bars=5 μm; (b) AFM height profiles.
Point 5: Overall, while the concept of research looks promising there is not enough data, and the presentation of the research in the manuscript is a bit confusing
Response 5: We believe the improvements in the text will further clarify the aim, the achievement and the potential impact of our research. Our scope is to show a new way to obtain something, already explored in literature, but in a more simple way. Unfortunately we are not able to provide other analysis at the moment, also due to the time constraints imposed by the review process. All the available data are already reported in the manuscript. However, further and in-depth analysis will be carried out in future studies, oriented to specific applications of the method we propose in the manuscript.
We would like to thank the referee again for taking the time to review our manuscript.

Reviewer 2 Report
This paper presented a novel technique to produce Ag nanoparticle. It sounds simple and easy to operate by an aqueous solution of silver nitrate and laser pulse. Effects of concentration and scanning velocity on the AgNP size, surface roughness and oxidation are investigated. There are some minor problems.
1. FIg. 6a shows a control line. What does it mean? You might explain this black line of FIg. 6a in the caption as you do for the black line in Fig. 6b.
2. The last sentence in the conclusion ‘The possibility ... makes the process bio-friendly and easy to scale up’, is not that clear. You might rewrite it with more details.
Author Response
Response to Reviewer 2 Comments
This paper presented a novel technique to produce Ag nanoparticle. It sounds simple and easy to operate by an aqueous solution of silver nitrate and laser pulse. Effects of concentration and scanning velocity on the AgNP size, surface roughness and oxidation are investigated. There are some minor problems.
We want to thank the reviewer for the time and the effort spent to assess our manuscript and for the valuable comments and suggestions provided.
Point 1: Fig. 6a shows a control line. What does it mean? You might explain this black line of Fig. 6a in the caption as you do for the black line in Fig. 6b.
Response 1: We apologize for the missing explanation in Figure 6 (now Figure 7). The missing information has been added to the caption:
Figure 7. Plot of surface roughness values (Sa) for each different combination of processing conditions: V=0.1, 0.2, 0.35, 0.5 mm/s and 0.01, 0.02, 0.1 wt.% AgNO3 concentration in solution. The black line represents the original roughness value of the untreated surface and its deviation (gray area).
Point 2: The last sentence in the conclusion ‘The possibility ... makes the process bio-friendly and easy to scale up’, is not that clear. You might rewrite it with more details.
Response 2: We agree with the reviewer that the final sentence in 4.Conclusion was simplistic, we believe that with the addition of more information and explanations it is now sufficiently clear. Please see the update paragraph below:
The possibility to achieve titanium structuring and oxidation, with no need for mechanical, chemical or thermal treatments, together with the synthesis and deposition of AgNPs makes our method viable for a variety of applications. Moreover the absence of additional chemicals(which can represent a source of pollution or contamination) and procedures, makes the process bio-friendly and easy to scale up. The disposal of hazardous waste and complex equipment and instrumentation, apart from the laser system, is not necessary.
We would like to thank the referee again for taking the time to review our manuscript.

Reviewer 3 Report
The modification of the titanium surface by the silver nanoparticles from AgNO3 aqueous solution using the femtosecond laser. Surface morphology and composition were studied by scanning electron microscope (SEM), X-ray diffraction (XRD), atomic force microscopy (AFM) and Raman spectroscopy. The effects of scanning speed and AgNO3 concentration on the surface properties were revealed and explained. The reported approach can be used for the design of antimicrobial surfaces on medical devices. The article is well written, text clear and easy to read, data is clearly presented, and the conclusions are supported by data. This work is interesting for broad range of readers and makes a significant contribution in materials sciences.
Several minor concerns are listed below:
1) Page 5: Based on the EDX data authors concluded that “Increasing the scanning speed, the amount of detected [O] increases” However, the difference in the Ti/O ratios for various scanning speed is within the experimental error (marked on the bars in Fig. 3a). I suggest to correct the statement.
2)Page 7: Authors concluded that, “a major contribution in the Raman spectra presented in both figures is expected to be due to titanium dioxide (TiO2) in the form of Anatase (characteristics peaks at 143, 397, 516, and 638 cm-1) and Rutile (characteristics peaks at 143, 241, 445 and 610 cm-1) [52].” For me, this observation is not obvious from the spectra in Fig 6. I suggest to enlarge this specific part of the spectra and assign Anatase and Rutile peaks on the spectra.
Author Response
Response to Reviewer 3 Comments
The modification of the titanium surface by the silver nanoparticles from AgNO3 aqueous solution using the femtosecond laser. Surface morphology and composition were studied by scanning electron microscope (SEM), X-ray diffraction (XRD), atomic force microscopy (AFM) and Raman spectroscopy. The effects of scanning speed and AgNO3 concentration on the surface properties were revealed and explained. The reported approach can be used for the design of antimicrobial surfaces on medical devices. The article is well written, text clear and easy to read, data is clearly presented, and the conclusions are supported by data. This work is interesting for broad range of readers and makes a significant contribution in materials sciences.
We want to thank the reviewer for the time and the effort spent to assess our manuscript and for the valuable comments and suggestions provided. We want to further thank the reviewer for the appreciation shown for our work.
First we want to apologize for the mistake in the abstract where XRD is mentioned in place of EDX, as could be noticed, such analysis is absent in our study.
The corrected sentence on page 1, lines 38-40 is reported below:
The processed surfaces were examined by means of scanning electron microscope (SEM), energy-dispersive X-ray spectroscopy (EDX) atomic force microscopy (AFM) and Raman spectroscopy
Several minor concerns are listed below:
Point 1: Page 5: Based on the EDX data authors concluded that “Increasing the scanning speed, the amount of detected [O] increases” However, the difference in the Ti/O ratios for various scanning speed is within the experimental error (marked on the bars in Fig. 3a). I suggest to correct the statement.
Response 1: The observation of the reviewer is correct, we agree in clarifying the interpretation of such data paying more attention to the experimental error. We have corrected the statement, on page 6 of the revised manuscript, lines 22-24
Increasing the scanning speed, the amount of detected [O] increases as well, which could be attributed to surface oxidation of the material.”
to
Increasing the scanning speed, the amount of detected [O] seems to increase as well, which could be attributed to further surface oxidation of the material. However even if the trend appears quite clear, the increase remains comparable to value deviation (see error bars).
We hope now the description is more accurate.
Point 2: Page 7: Authors concluded that, “a major contribution in the Raman spectra presented in both figures is expected to be due to titanium dioxide (TiO2) in the form of Anatase (characteristics peaks at 143, 397, 516, and 638 cm-1) and Rutile (characteristics peaks at 143, 241, 445 and 610 cm-1) [52].” For me, this observation is not obvious from the spectra in Fig 6. I suggest to enlarge this specific part of the spectra and assign Anatase and Rutile peaks on the spectra.
Response 2: We agree with the reviewer that the Raman spectra are not providing enough information to identify which kind of oxide is present. We recognize, in agreement with the reviewer, that oxide identification was incorrect. Results from literature (now cited in the manuscript) indicate amorphous TiO2 as the main component of the oxide layer. However other contributions cannot be excluded (e.g. Ti2O3). The text has been corrected accordingly. Please see the corrected paragraph and relative reference:
According to Landis et al.[60], a major contribution in the Raman spectra presented in both figures is expected to be due to titanium dioxide (TiO2) in a low crystallinity to amorphous state; the characteristic peaks of Anatase and Rutile cannot be identified. However the presence of different oxides, such as Ti2O3 cannot be excluded.
- Landis, E.C.; Phillips, K.C.; Mazur, E.; Friend, C.M. Formation of nanostructured TiO2 by femtosecond laser irradiation of titanium in O2. J. Appl. Phys. 2012, 112, 063108.https://doi.org/10.1063/1.4752276
In light of this we believe that adding peak assignment on the plot for a few different possible oxides, would result in a confusing picture, considering also that different oxides have characteristic peaks overlapping each other.
We would like to thank the referee again for taking the time to review our manuscript.
We are also glad to know that the reviewer found our study interesting and valid.

Reviewer 4 Report
In this work, authors performed micro and nano structuring of commercially pure titanium (cp-Ti) immersed in aqueous solution with varying weight percentages of AgNO3. Authors performed optimization studies of laser ablation with varying scanning speeds analyzed the thus formed micro-nano structures with different characterization techniques such as scanning electron microscope (SEM), X-ray diffraction (XRD), atomic force microscopy (AFM) and Raman spectroscopy. Overall, the work performed by the authors is systematic but however the scope of this manuscript could be improved by performing (a) more or repeating the experiments and characterizations, and (b) some more application-oriented studies demonstrating the capabilities of the fabricated Ti Nano-micro structures. There are only a few to none grammatical mistakes, according to me, hence the article looks good coming to the English as well presentation point of view.
However, authors are requested to consider these minor revisions for their future submissions.
1. Page 3: Authors indicated that the liquid level above the target is 1 mm. This level of liquid is very less and usullay researchers immerse the targets and ensure a height of use 5-10 mm. Use of minimum height of liquid will result in improper laser ablation (or laser0matter interaction itself) as the liquid itself gets evaporated even before the laser energy used for ablation. Hence the experiments should be performed again with suitable liquid level above the target.
2. Figure 2: Authors provided SEM images of the obtained targets in the laser ablation studies. The authors should have analyzed the pure NPs of Ti generated in the studies too.
3. Did authors characterized the NPs generated in the laser ablation of Ti with different wt% of AgNO3? The formed NPs could be of bimetallic NPs with varying structures such as core-shell NPs, amalgamated NPs, etc. It would be interesting to see the morphological and elemental composition of the formed NPs in each condition.
4. The fabricated structures have Ag NPs on them. Any application of these structures towards using these as SERS substrates for detecting common dyes or any chemicals or any bactericidal activity. This substrates could be used as nanoparticle enhanced laser induced breakdown spectroscopy (NE-LIBS) targets as well by demonstrating some enhancement in LIBS signal of a analyte on pure Ti to that of Ag@Ti nano-micro structures.
Author Response
Response to Reviewer 4 Comments
In this work, authors performed micro and nano structuring of commercially pure titanium (cp-Ti) immersed in aqueous solution with varying weight percentages of AgNO3. Authors performed optimization studies of laser ablation with varying scanning speeds analyzed the thus formed micro-nano structures with different characterization techniques such as scanning electron microscope (SEM), X-ray diffraction (XRD), atomic force microscopy (AFM) and Raman spectroscopy. Overall, the work performed by the authors is systematic but however the scope of this manuscript could be improved by performing (a) more or repeating the experiments and characterizations, and (b) some more application-oriented studies demonstrating the capabilities of the fabricated Ti Nano-micro structures. There are only a few to none grammatical mistakes, according to me, hence the article looks good coming to the English as well presentation point of view.
We want to thank the reviewer for the time and the effort spent to assess our manuscript and for the valuable comments and suggestions provided.
First we want to apologize for the mistake in the abstract where XRD is mentioned in place of EDX, as could be noticed, such analysis is absent in our study. The corrected sentence on page 1, lines 38-40 is reported below:
The processed surfaces were examined by means of scanning electron microscope (SEM), energy-dispersive X-ray spectroscopy (EDX) atomic force microscopy (AFM) and Raman spectroscopy.
We agree with the reviewer that performing more experiments and application oriented studies would enrich our work, however this goes beyond the scope of our paper. Furthermore, there is no other data available at the moment and also due to the limited time for the review process, we are unable to make such additions.
However, authors are requested to consider these minor revisions for their future submissions.
Point 1: Page 3: Authors indicated that the liquid level above the target is 1 mm. This level of liquid is very less and usullay researchers immerse the targets and ensure a height of use 5-10 mm. Use of minimum height of liquid will result in improper laser ablation (or laser0matter interaction itself) as the liquid itself gets evaporated even before the laser energy used for ablation. Hence the experiments should be performed again with suitable liquid level above the target.
Response 1: We appreciate the suggestion of the reviewer, and we will take it in account for our future experiments. We want to thank you for your comment also because you pointed out an error, made by writing these parameters, for which we apologize. Indeed, we had a 2 mm thick sample immersed in 5 mm of solution, resulting in 3 mm of liquid above the target. Due to that, we believe that the scenario described by the reviewer is not likely to occur. Moreover, the liquid (H2O) is transparent to the used wavelength (800 nm), so direct evaporation could happen only in proximity of the focal spot (a few tens of microns in diameter), that is orders of magnitude smaller than the liquid layer thickness. Moreover the container (cylindrical with a ~50 mm diameter) is much larger than the sample (10x10 mm2 with a 2 mm thickness), so the total amount of water is enough to remain constant during the whole process.
Such details have been added in Material and Methods and the mistakes corrected. Please see below the updated paragraph, page 4, lines 13-19 and 26-27 :
Commercially pure Titanium plates (Sigma Aldrich, 10 mm x 10 mm x 2 mm), placed in a cylindrical container (with a 55 mm diameter) and immersed in an aqueous solution with rising concentrations (0.0, 0.6, 1.2 and 6 mM or 0.0, 0.01, 0.02, 0.1 wt.%) of Silver Nitrate (AgNO3), were patterned using a Ti:Sapphire laser (Quantronix-Integra-C, Hamden, CT, USA) with pulse duration τ < 130 fs, central wavelength λ = 800 nm and repetition rate ω = 1 kHz. The experimental setup and condition are schematically shown in Figure 2.
The laser beam, 7 mm in diameter, was focused on the sample surface, through 3 mm of solution, by an achromatic biconvex lens with a 100 mm focal distance.
Point 2: Figure 2: Authors provided SEM images of the obtained targets in the laser ablation studies. The authors should have analyzed the pure NPs of Ti generated in the studies too.
Response 2: We reported the presence of Ti NPs but we have not performed in-depth analysis, because their distribution on the surface is quite sparse, representing a redeposit of ablated materials.
Point 3: Did authors characterized the NPs generated in the laser ablation of Ti with different wt% of AgNO3? The formed NPs could be of bimetallic NPs with varying structures such as core-shell NPs, amalgamated NPs, etc. It would be interesting to see the morphological and elemental composition of the formed NPs in each condition.
Response 3: The possibility to have bimetallic NPs appears intriguing for us also, however EDX analysis has shown only Ti and O signals coming from such NPs. The inclusion of Ag atoms cannot be excluded, however the Ti NPs have shown no clear signal of silver in any experimental condition. It is also true that the size of such NPs, generally below 500 nm in diameter, and their scarcity on the surface, makes them difficult to be further analyzed.
Point 4: The fabricated structures have Ag NPs on them. Any application of these structures towards using these as SERS substrates for detecting common dyes or any chemicals or any bactericidal activity. This substrates could be used as nanoparticle enhanced laser induced breakdown spectroscopy (NE-LIBS) targets as well by demonstrating some enhancement in LIBS signal of a analyte on pure Ti to that of Ag@Ti nano-micro structures.
Response 4: We are truly grateful to the reviewer to have suggested another possible application for our method. NE-LIBS as well as other applications in plasmonics are now mentioned in the final part of 3.Results and Discussion. Please see page 10 of the revised manuscript, lines 15-23,
Similar structures could also find application for Laser Induced Breakdown Spectroscopy (LIBS), especially in Nanoparticle Enhanced LIBS (NELIBS) [72] which still represents a developing field [73]. Surface patterning with metal NPs allows also the exploitation of thermoplasmonic effect, opening another wide range of potential applications [74,75] for our method which provide, in addition, surface structuring. However, finer control in NPs size and distribution is required to ensure an optimal match between NPs plasmonic resonance and the used laser wavelength, necessary condition for the aforementioned applications. Further experiments and investigations will explore the potentiality of our method for such applications as well.
Unfortunately, as mentioned before, we were unable to test the material processed with our method for specific applications, as this would need specific design for each specific application.
We believe the improvements in the text will further clarify the potential impact of our research.
We would like to thank the referee again for taking the time to review our manuscript.

Round 2
Reviewer 1 Report
After reviewing the responses from the authors, I feel all issues have been appropriately addressed and I recommend the manuscript be accepted for publication.
Reviewer 4 Report
Authors revised the MS according to the reviewer' suggestion. The revised MS can be accepted for publication